# Digital health and the promise of equity in maternity care: A mixed methods multi-country assessment on the use of information and communication technologies in healthcare facilities in Latin America and the Caribbean

Ariadna Capasso[1,2]*, Mercedes Colomar[3], Dora Ramírez[4], Suzanne Serruya[3], Bremen de Mucio[3]

1 Health Resources in Action, Boston, MA, United States of America, 2 New York University, New York, NY, United States of America, 3 Pan American Health Organization/World Health Organization, Montevideo, Uruguay, 4 Independent Consultant, Asunción, Paraguay

* acapasso@hria.org

**Data Availability Statement:** The data underlying the results presented in the study are available on

## Abstract

### Introduction

Timely access to maternity care is critical to saving lives. Digital health may serve to bridge the care chasm and advance health equity. Conducted in the aftermath of the COVID-19 pandemic, this cross-sectional mixed-methods study assessed the use of information and communication technologies (ICTs) in healthcare facilities in nine Latin American and Caribbean countries to understand the landscape of ICT use in maternity care and the barriers and facilitators to its adoption.

### Materials and methods

Between April 2021 and September 2022, we disseminated an online survey in English and Spanish among, mainly public, healthcare institutions that provided maternity care in Argentina, Bolivia, Colombia, the Dominican Republic, Ecuador, Guyana, Honduras, Paraguay and Peru. We also interviewed 27 administrators and providers in ministries of health and healthcare institutions.

### Results

Most of the 1877 institutions that answered the survey reported using ICTs in maternity care (N = 1536, 82%), ranging from 96% in Peru to 64% in the Dominican Republic. Of institutions that used ICTs, 59% reported using them more than before or for the first time since the pandemic began. ICTs were most commonly used to provide family planning (64%) and breastfeeding (58%) counseling, mainly by phone (82%). At the facility level, availability of equipment and internet coverage, coupled with skilled human resources, were the main

OpenICPSR: https://www.openicpsr.org/openicpsr/
project/195667/version/V1/view.

**Funding:** "This study was funded by Global Affairs
Canada through an agreement with the Pan
American Health Organization/the World Health
Organization. Global Affairs Canada was not
involved in the study design, collection, analysis,
and interpretation of data, writing, or submission."

**Competing interests:** The authors have declared
that no competing interests exist.

factors associated with ICT use. At country level, government-led initiatives to develop digital health platforms, alongside national investments in the digital infrastructure, were the determining factors in the adoption of ICTs in healthcare provision.

## Conclusion

Digital health for maternity care provision relied on commonly available technology and did not necessitate highly sophisticated systems, making it a sustainable and replicable strategy. However, disparities in access to digital health remain and many facilities in rural and remote areas lacked connectivity. Use of ICTs in maternity care depended on countries' long-term commitments to achieving universal health and digital coverage.

## Introduction

Maternal mortality is a grave public health problem in Latin America and the Caribbean, where an estimated 8400 women died in 2020 from causes related to pregnancy and childbirth [1]. It is the only region in the world that has shown no significant improvements in maternal mortality reduction since 2000 [1]. Further, the slow progress made towards reducing maternal deaths over the last two decades was undone by the COVID-19 pandemic: the maternal mortality ratio increased from 77 deaths per 100,000 live births in 2019 to an estimated 88 deaths per 100,000 live births in 2020 [1].

Deep-seated structural inequalities contribute to disparities in maternal deaths within and between countries [1, 2]. Women of low socioeconomic status, living in rural and remote areas, and of indigenous and African descent, bear the burden of maternal mortality [2]. The maternal mortality ratio of indigenous and afrodescendent women was 1.5 to 3 times higher than that of women of other race/ethnicities every year in Brazil, Guatemala, and Paraguay between 2015 and 2021 [2]. In Paraguay, the maternal mortality ratio among women who did not attend school was 5 times higher than among those with a graduate degree in 2020 [2]. A study examining deaths related to pregnancy and childbirth associated with COVID-19 found that 92% of deaths were among non-White women and 33% among those who had not finished secondary school [3]. The tragedy is that most maternal deaths are preventable with universal access to sexual and reproductive care [4].

Digital health can be a powerful tool to improve population health in resource-constrained settings. In 2005, the World Health Organization (WHO) member states embraced the adoption of digital health as a strategy to bridge the healthcare access gap and attain universal health coverage [5]. WHO defines digital health as "the cost-effective and safe use of information and communication technologies in support of health and health-related areas," stating that strengthening health systems through digital health "reinforces fundamental human rights by improving equity, solidarity, quality of life, and quality of care" [6]. In 2011, building on this commitment and under the umbrella of the Pan American Health Organization (PAHO), the countries of Latin America and the Caribbean adopted a plan of action to accelerate progress towards attaining universal health coverage and reducing inequities in access to care through the integration of information and communication technologies (ICTs) in healthcare systems and services [7].

Digital health strategies have two components: the technology used and the main purpose for its use. With this in mind, digital health encompasses a wide variety of strategies that incorporate ICTs to provide and obtain health-related information and advice when distance is a

factor, including the remote delivery of healthcare services by providers to patients, virtual consultations among healthcare providers, access to health information by patients (e.g., a health application, an online platform, or a phone line), storage and management of health data (e.g., electronic medical records), telemonitoring through devices that track patient health information at a distance, and management of scheduling and reminders [8]. Digital health interventions in maternity care include virtual routine antenatal and postpartum care visits to provide health counseling, identify warning signs, and answer patient questions; wearables and applications for telemonitoring of pregnant patients with existing conditions, such as gestational diabetes and hypertension; breastfeeding, sexual and reproductive health, and mental health counseling by phone or video call; using SMS and/or e-mails to send appointment reminders and health information to pregnant and postpartum women; remote consultations between primary care and specialty providers to resolve complex cases; and maternal health data management and surveillance through platforms such as PAHO's Perinatal Information System, among others.

The last PAHO/WHO digital health survey, conducted in 2016, revealed that countries in Latin America and the Caribbean had made significant progress in the area. For example, 61% of countries had a national digital health policy; 90% used teleradiology to store and share radiological images for diagnosis or consultation; 58% used telemonitoring to track patients' vitals at home, particularly for patients with chronic conditions; and 100% employed social media for health education [9]. However, the survey, completed by health ministries, did not inquire about the use of ICTs in maternity care, specifically, nor on the specificities of types of technologies used and services provided with the assistance of ICTs. Further, this data are likely outdated given the rapid penetration of digital technologies in all sectors, particularly since the COVID-19 pandemic [10]. For example, a global systematic review found significant decreases in in-person and increases in virtual antenatal care visits during the COVID-19 pandemic [11]. A 2020 global survey of healthcare providers found that 24% of those in middle income countries used some ICT in maternity care during the COVID-19 pandemic [12].

The evidence on digital health in maternity care is mixed. A review of interventions in high-income countries found that digital health interventions that supplemented in-person mental health and diabetes care during pregnancy with remote contacts resulted in similar or improved clinical outcomes [13]. Further, a reduced-visit antenatal care schedule using telehealth to replace routine in-person visits in the U.S. obtained similar clinical outcomes and higher patient satisfaction than usual care [14]. A systematic review of ICT use in sexual and reproductive health services in Latin America and the Caribbean found that most interventions entailed the use of SMS for reminders or social media and web-based platforms for education, but fewer involved direct service provision [15]. A caveat of this study, however, was that, since the majority of interventions carried out in health facilities across Latin America and the Caribbean are not documented in the literature, a literature review provides only a partial picture of the reality on the ground. The following are two examples of evaluated digital health interventions in the region. A hybrid antenatal care model for women with high-risk pregnancies in Peru appeared feasible, but no outcomes were reported [16]. In Colombia, an intervention to strengthen the capacity of rural hospitals to respond to obstetric emergencies entailing telehealth and distance learning resulted in a 29% reduction in perinatal deaths [17].

Given the rapid growth of digital health in the aftermath of COVID-19 and the limited evidence on the extent of ICT integration in maternity care, this study aimed to:

1. Describe the prevalence of ICT use in maternity care, the types of ICTs used, and the services provided using ICTs in a sample of health facilities in 9 countries of Latin America and the Caribbean,

2. Understand changes in the use of ICTs as a result of the COVID-19 pandemic, and

3. Identify the main factors associated with ICT use in maternity care.

## Materials and methods

This is a cross-sectional mixed methods study of health facilities providing maternity care in 9 Latin American and Caribbean countries that participated in a PAHO-led program to improve primary care access: Argentina, Bolivia, Colombia, the Dominican Republic, Ecuador, Guyana, Honduras, Paraguay and Peru. The study included health facilities that provided maternity care in the countries under consideration, and excluded those that did not provide such care. Quantitative data was collected with an online survey questionnaire and qualitative data via key informant interviews, as detailed below. Survey data served to primarily answer questions related to the prevalence of ICT use, the types of ICTs used, and the services provided using ICTs, whereas qualitative data served to gain nuanced information about the context in which ICTs were employed, and about the barriers and facilitators to their adoption. Data for 8 countries were collected between April and September 2021, and for Colombia between July and September 2022. The New York University Institutional Review Board (IRB) did not consider the present study human subjects research because the information collected was not about individuals but about methods, policies, procedures, and organizations; thus, the study did not require IRB submission.

### Survey sampling

The survey sampling frame varied by country and was based on what was deemed feasible by national partners (See S1 Table. Sampling frame by country). Briefly, the Dominican Republic, Guyana, Honduras, and Peru considered a national sample of all facilities; Argentina and Bolivia considered a national sample of maternity hospitals and facilities associated with the Department of Telemedicine of the Ministry of Health, respectively; and Colombia, Ecuador, and Paraguay focused on priority regions. The country teams decided whether to limit the survey to the public sector or not. Most countries focused on public sector facilities, while Guyana, Paraguay, and Peru included those in both the public and private sectors. With the help of national partners, including ministries of health and PAHO country offices, an invitation to complete the survey with a link to an online questionnaire was distributed via e-mail and WhatsApp to maternity health unit directors or telehealth focal points. Instructions asked the potential respondents to complete one survey per facility and to obtain information from other departments, as needed. When the contact information was available, telephone follow-up was conducted to non-responders a week after the initial invitation was sent. Respondents provided informed consent online by checking two boxes affirming that they understood the terms of the survey and that their participation was voluntary.

### Survey questionnaire

A survey questionnaire was developed in Spanish by an international team of public health practitioners, researchers, and providers with expertise in maternal health and digital health. The questionnaire was informed by PAHO's conceptual framework for telemedicine implementation [8] and guided by the global eHealth survey [9]. The questions were organized under the following themes: 1) general information about the institution and the population served; 2) whether ICTs were used, types of ICTs used, if any, and services provided using ICTs; and 3) enablers of ICT use. To gather data on point 3, respondents were offered a list of five social and nine institutional factors commonly associated with ICT use and asked to rate

the extent to which each facilitated their adoption. These factors included budget appropriations, and support to providers, such as training and reimbursement for expenditures incurred when using personal devices. ICT use was assessed by the question "Does your facility use any information and communication technology (ICT) tools in maternity care (e.g., phone calls, video calls, telemonitoring, text messaging, etc.)?" (Answer options: Yes, the same way as we did before the onset of the pandemic; Yes, more than before the pandemic; Yes, we started using ICT tools when the pandemic was already under way; and No). The variable was dichotomized (Yes/No). The questionnaire was piloted with maternity care providers and administrators in each country to assess face validity and tailor the wording of questions and response options to each national context. The questionnaire was then translated into English and programmed in both languages on Qualtrics (Provo, UT, USA) (See S2 Text. Online survey questionnaire).

## Quantitative analysis plan

Quantitative analysis involved producing descriptive statistics (frequencies and percentages) using Stata version 17.0 (StataCorp, College Station, TX, USA). Descriptive statistics revealed the over-representation of facilities from Peru and Ecuador. A sensitivity analysis was conducted, showing that in Ecuador and Peru 70% and 96% of respondents used digital health, respectively, compared to 82% on average across all countries. We did not apply any statistical corrections as sampling strategies varied by country.

## Qualitative data collection and analysis

In the 7 countries where we found promising examples of ICT use in maternity care, key informant interviews were conducted with maternal health and telehealth focal points in health ministries, and with maternity care providers, to learn more about these interventions. Consent was obtained orally prior to beginning the interview. Overall, 27 in-depth interviews were conducted, lasting from 40 to 60 minutes each. The key informant interview guide consisted of four sections to better understand the digital health strategies implemented and the factors associated with implementation including: 1) details about any digital health intervention used; 2) funding to develop and implement the intervention; 3) enablers of implementation; and 4) lessons learned and future plans for digital health (See S1 Text. Interview guide).

The information gathered from the different data collection methods was complementary: whereas the survey allowed us to collect quantitative data from many healthcare facilities, the qualitative data provided us nuanced information about specific interventions and their implementation context. The qualitative analysis was conducted in NVivo 14 (Lumivero, Denver, CO, USA). We employed a conventional content analysis approach to identify key contextual factors related to utilization of ICTs [18]. Quantitative and qualitative analyses were done simultaneously in a concurrent design.

## Secondary data

To better understand the setting in which the digital health intervention was implemented, we explored macro factors that could affect the ability of facilities to use ICTs. We examined national level factors related to the digital infrastructure, access, and affordability of ICTs, and a general economic marker. Digital environment data are drawn from the International Telecommunication Union (ITU) Digital Development Dashboard [19]; and the country economies are based on the 2022–2023 World Bank classifications [20]. Lower-middle-income countries are those with a per capita Gross National Income (GNI) of $1,086-$4,255 and upper-middle-income countries are those with a per capita GNI of $4,256-$13,205. For

purposes of visualization, the Infrastructure and Access values below the regional average are marked in red on the table. The Broadband Commission for Sustainable Development established as a 2025 target that broadband services be affordable in developing countries, at a cost of less than 2% of the monthly per capita GNI [21]. Affordability values above 2% GNI, the global affordability threshold, are marked in red.

## Results

### Sample characteristics

After removing 24 duplicates, we analyzed 1877 survey responses. Duplicates occurred when respondents continued working on unsubmitted surveys from a different device. The mean response rate was 71% (95% Confidence Interval: 44%-97%) (See S2 Table. Response rate by country). Three-quarters (75%) of respondents completed 100% of the survey, and the rest were partial completers.

Primary health care centers (53%) and health posts (23%) comprised the majority of the sample (Table 1). Consistent with this, over three-quarters of institutions represented were primary care level institutions (76%). Peru (23%) and Ecuador (21%) made the largest proportion of the sample, followed by Bolivia (16%).

### Use of ICTs

Overall, 82% (n = 1536) of health facilities reported using ICTs in maternity care (Fig 1). Peru was the country with the highest proportion of facilities reporting ICT use (96%), followed by

**Table 1. Use of information and communication technologies (ICTs) in the provision of maternity care in selected countries of Latin America and the Caribbean, by type of institution and country (2021–2022), based on survey responses.**

| Variable | Total | Used ICTs |
|---|---|---|
| | n (%) | |
| | 1877 (100.0) | 1536 (81.8) |
| **Institution type** | | |
| Referral hospital | 160 (8.6) | 150 (9.8) |
| District/regional hospital | 180 (9.7) | 127 (8.3) |
| Primary health care center or clinic | 990 (53.1) | 794 (52.0) |
| Health post | 422 (22.6) | 360 (23.6) |
| Administrative center/health network | 113 (6.1) | 96 (6.3) |
| **Level of care** | | |
| Primary | 1440 (78.1) | 1175 (77.8) |
| Secondary | 230 (12.5) | 176 (11.7) |
| Tertiary | 145 (7.9) | 135 (8.9) |
| **Country** | | |
| Argentina | 73 (3.9) | 56 (3.7) |
| Bolivia | 305 (16.3) | 254 (16.5) |
| Colombia | 212 (11.3) | 183 (11.9) |
| Dominican Republic | 39 (2.1) | 25 (1.6) |
| Ecuador | 402 (21.4) | 277 (18.0) |
| Guyana | 72 (3.8) | 50 (3.3) |
| Honduras | 83 (4.4) | 58 (3.8) |
| Paraguay | 260 (13.9) | 219 (14.3) |
| Peru | 431 (23.0) | 414 (27.0) |

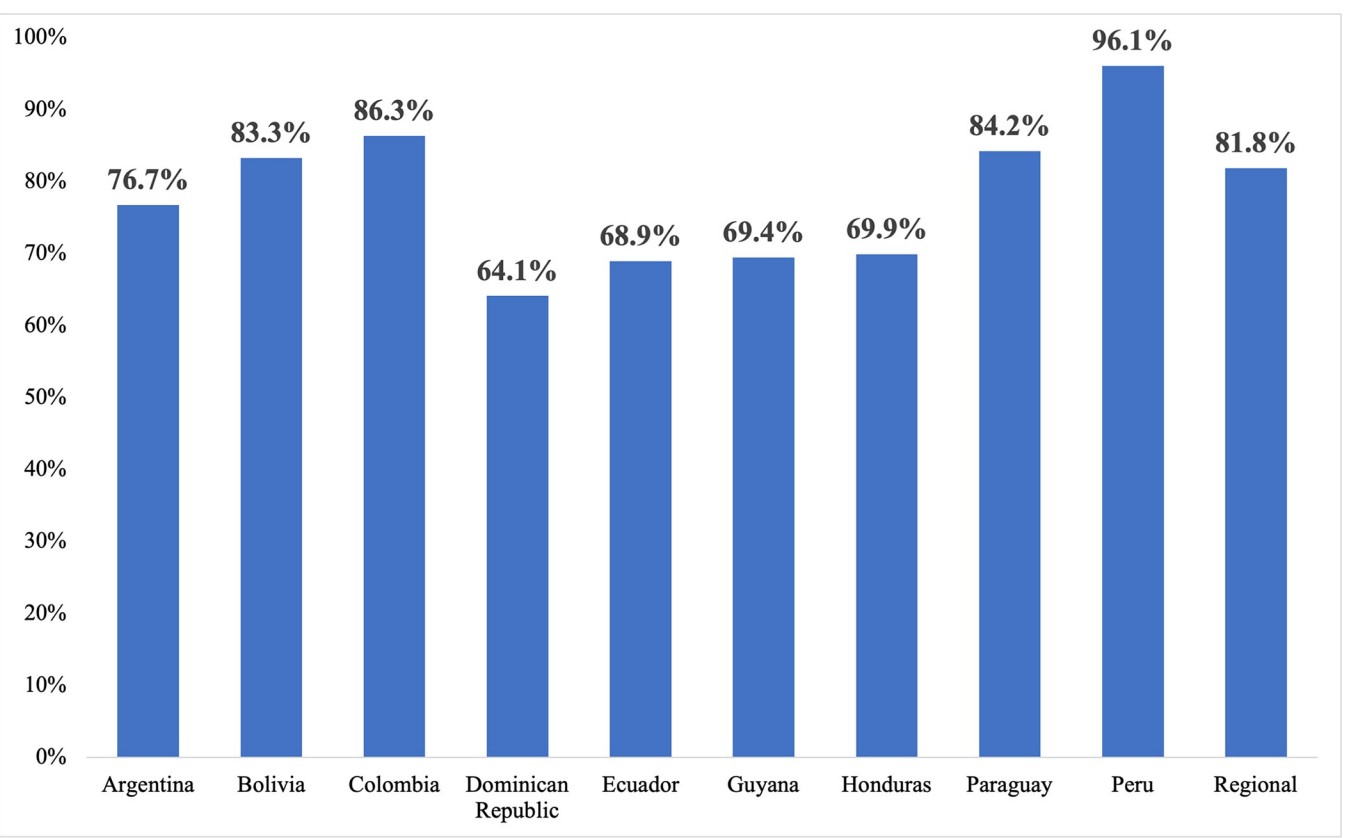

**Fig 1. Proportion of institutions that used ICTs, by country (2021–2022), based on survey responses (N = 1877).**

Colombia (86%). The Dominican Republic was the country with the lowest proportion of facilities reporting ICT use (64%). When examining ICT use by institution type and care level, the highest proportion of ICT use was indicated by referral hospitals (94%) and tertiary care level centers (93%), respectively. In comparison, 80% of primary health care centers used ICTs.

Family planning (64%) and breastfeeding (58%) counseling were the services most commonly provided through the use of ICTs (Table 2). The services less frequently offered via ICT were safe abortion counseling (9%), medical prescription orders (24%), and telemonitoring or long-distance monitoring of high-risk conditions using applications or digital devices (25%). Of note, the proportion of facilities that reported ICT use to provide safe abortion counseling was higher in countries where the procedure is legal in all or certain circumstances, Argentina (36%) and Colombia (30%), and lowest where it is highly criminalized, such as Honduras (0%) (See S3 Table. Use of ICTs for abortion counseling, by country). Other services mentioned included teleconsultations between professionals, management of domestic violence situations, management and treatment of sexually transmitted infections, and care of pregnant women with COVID-19 infection, among others.

In terms of the ICTs used in maternity care, an overwhelming majority of facilities used the phone (82%), including applications that enable calling, to communicate with clients (Table 2). To a much lesser extent, other ICTs used included electronic medical records to access client health data (27%), a hotline or dedicated phone number to address pregnancy concerns (27%), and video calls for telemedicine consultations (27%). Provision of telemonitoring devices to pregnant women and engaging in remote recording and monitoring of

**Table 2. Maternity care and information and communication technologies (ICTs) in nine Latin American and Caribbean countries (2021–2022), based on survey responses.**

| Variable | Proportion N = 1494 (100.0)* |
|---|---|
| **Types of services provided using ICTs** | |
| Safe abortion counseling | 134 (9.0) |
| Orders for medical prescriptions | 361 (24.2) |
| Telemonitoring | 374 (25.0) |
| Gestational disease management | 459 (30.7) |
| Diagnostic antenatal care visit | 571 (38.2) |
| Birth preparation | 585 (39.2) |
| Routine postnatal care visit | 628 (42.0) |
| Routine antenatal care visit | 779 (52.3) |
| Breastfeeding counseling | 867 (58.0) |
| Family planning counseling | 962 (64.4) |
| Other | 357 (23.9) |
| **ICTs used**** | |
| Telemonitoring applications | 88 (6.0) |
| Email address for pregnancy concerns | 92 (6.2) |
| Provision of phones or data to pregnant women | 118 (8.0) |
| Provision of telemonitoring devices to pregnant women | 122 (8.3) |
| Web-based platform for data sharing | 185 (12.6) |
| mHealth (text messages) | 217 (14.7) |
| Video calls for telemedicine consultations | 392 (26.6) |
| Hotline to address pregnancy concerns | 397 (26.9) |
| Electronic medical records | 404 (27.4) |
| Phone calls, including via Apps, for consultations | 1205 (81.8) |
| Other | 133 (9.0) |
| **Protocols and data security**** | |
| Telehealth protocol in place | 378 (27.7) |
| Data security system | 664 (49.6) |

*Note*: *Does not add to 100 because multiple options were possible

** n = 1,474 due to missing

*** n = 1363 and 1339, respectively, due to missing

pregnancy-related conditions, such as blood sugar levels or blood pressure measurements, by providers were not commonly used. Other responses included using social media (such as Facebook), and national health information platforms (such as WawaRed in Peru [22] or the Federal Platform for Telehealth and Distance Communication in Argentina [23]). Digital health norms and protocols are important to ensure quality of telecare and compliance with ethical standards. Among respondents, 38% of institutions had a telehealth protocol in place and 50% had systems to ensure data security.

Interviewees offered more nuanced descriptions of how ICTs were used to provide services to women during pregnancy and the puerperium. Interviewees described using ICTs to schedule appointments, to conduct routine antenatal visits, for remote gestational disease management, and for medication prescribing. In Colombia, national guidelines during COVID-19 recommended antenatal teleconsultations to conduct anamnesis, maternal risk assessment and monitoring, behavioral counseling, and laboratory analysis requests [24]. Protocols of a tertiary maternity hospital in Peru for remote antenatal care visits recommended similar

services, but also included orders for imaging, as relevant to gestational age, as well as COVID-19 tracing [16]. Interviewees in Peru considered having electronic medical records, which providers could access from home, if needed, as a key enabler of teleconsultations.

In addition to direct services, an interviewee from the Ministry of Health of Bolivia, considered ICTs critical to coordinate obstetric emergency responses. He described that staff in referral and emergency coordinating centers in each region used cell phones to coordinate the response to obstetric emergencies and referrals, including ambulance dispatch. Other interviewees from Bolivia and Paraguay mentioned the development of geospatial analysis tools to inform reproductive health supply chain management in remote areas based on the numbers of reproductive-aged and pregnant women. In both cases, the development of these tools was funded by international development and tested as pilot trials. They were complex and resource intensive interventions and, whereas promising in small scale implementation, they proved difficult to sustain, scale-up, and integrate into public healthcare systems.

Interviewees from Argentina, Colombia, Peru, and Paraguay mentioned the phone as the most common ICT used to reach pregnant women, both for phone calls and text messaging. In Peru and Paraguay, nurse midwives mentioned forming WhatsApp groups of pregnant women to provide healthy behavior recommendations and to answer questions. A midwife in Paraguay explained, "*each midwife had her own list of patients. [Providers] became more accessible to clients by using WhatsApp to answer concerns and to discuss medical lab results.*" Interviewees described how their facility used social media tools (such as Facebook) for risk communications, to share facility updates, to provide general health recommendations to pregnant women (e.g., vaccination reminders), and for event announcements (e.g., birth preparation workshops). Interviewees in Peru and Argentina described using ICTs, specifically Zoom and YouTube, to offer birth preparation classes. A midwife in Peru recounted opening an educational YouTube channel for pregnant women and recent mothers featuring different specialists, such as pediatricians, endocrinologists, and nutritionists, among others.

## Factors associated with ICT use

In terms of strategies implemented by institutions to promote ICT use, overall, 45% of facilities indicated providing some support to staff (Table 3). Of these, 46% trained staff on ICT use, either through in-house programs or by collaborating with health authorities. Only a small percentage of facilities (6%) reimbursed providers for costs incurred when using personal

**Table 3. Staff support and funding to support ICT use in maternity care among facilities that used ICTs, based on survey results, 2021–2022 (n = 1391).**

| Staff support | n (%) |
|---|---|
| Facility supported staff | 625 (44.9) |
| Trained providers in ICT use | 643 (46.2) |
| Reimbursed personal internet costs* | 39 (6.2) |
| Distributed cell phones to providers* | 103 (16.5) |
| IT staff assigned to solve problems* | 147 (23.5) |
| Designated telehealth focal point* | 378 (60.5) |
| Institution received funding for telehealth | 117 (8.7) |
| During the pandemic** | 65 (55.6) |
| Public funds** | 98 (83.8) |

*Percentage calculated over number of institutions that provided support (n = 625); **Percentage calculated over those who received funding (n = 117)

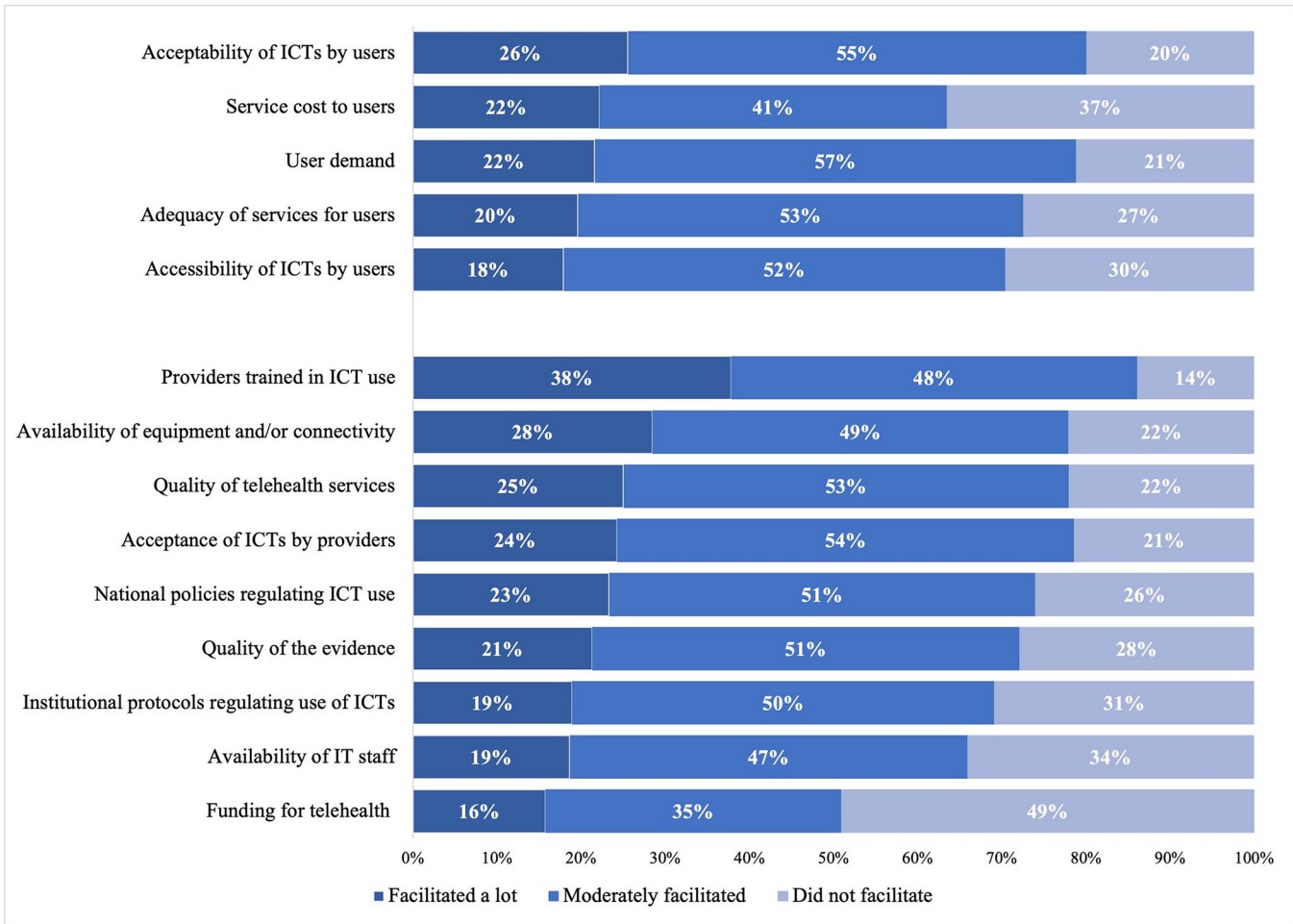

**Fig 2. Social and institutional factors associated with ICT use among institutions that used ICTs, by level of importance.**

devices or the internet, and 16% furnished providers with cell phones or other mobile devices for telehealth. Roughly 24% of facilities had dedicated IT personnel to address any technological issues encountered while using ICTs, and 60% had a designated telehealth focal point who served as a point of contact for coordinating telehealth use and addressing provider inquiries.

Respondents perceived ICT acceptability by users (26%) as the strongest facilitating social factor. Conversely, cost (37%) and accessibility of ICTs (30%) were the two factors perceived to least facilitate ICT adoption ([Fig 2]). In terms of institutional factors, 38% of respondents indicated that providers trained in the use of ICTs, and 28% that the availability of adequate equipment and technology, were the main facilitators of ICT adoption. Conversely, 49% of respondents perceived that funding for telehealth was not a facilitating factor. This is not surprising, as only 9% of institutions reported receiving such funding. These digital health initiatives were mostly funded by the public sector (83%). Among these, over half (56%) received the funding during the COVID-19 pandemic.

Qualitative data provided more context to the adoption of ICTs. There were mixed responses related to willingness of providers and patients to use ICTs. Some interviewees mentioned that providers and users were open to using ICTs, particularly cell phones which are ubiquitous in many Latin American contexts, to provide and receive services. An interviewee from Bolivia explained, *"[using ICTs] is acceptable to many women by telephone. And it is*

*accessible. Many users have cell phones with WhatsApp."* An interviewee from Paraguay noted that communicating with providers by message, calls, and videocalls was acceptable to users but, *"it's very precarious, there are few phones available, and the lines become very slow."* However, other interviewees mentioned that providers were reticent to adopt new technologies in their practice. One interviewee described, *"The obstetricians were afraid of giving advice on Zoom. We had to offer a tutorial on the ABC of using Zoom to get them to participate."* In this sense, several interviewees mentioned that capacity building on the use of ICTs, including on protocols for their use, and having IT staff available to solve technological issues, were necessary to promote their adoption. A barrier mentioned by an interviewee from Bolivia was the lack of computer science engineers working at the Ministry of Health. Some interviewees explained that many low-income households access internet via a cell phone, the only digital device at home. Thus, to receive phone calls or participate in online sessions, women had to negotiate the use of the cell phone with their partners, and even their children. One interviewee described, *"We had to really work on empowering women to talk to their husbands and convince them that it was important to leave the cell phone with internet at home."*

According to interviewees, having the digital infrastructure in place was a determinant of ICT use. In Argentina and Peru, for example, the government has invested in building the digital infrastructure for decades. Most public facilities are equipped with internet connection and IT equipment, such as computers. Instead, in Guyana, Honduras, and Paraguay, interviewees mentioned that some regions do not have internet coverage and that many primary care facilities in the public sector lack basic digital infrastructure. A midwife from Paraguay, for example, mentioned staff having to pay out-of-pocket for the internet at work. She explained, *"A group of 22 nurse midwives came together to pay for the monthly internet fee, this allows us to upload productivity, use email, connect via WhatsApp."* One of the biggest challenges reported were the areas with poor internet connection and intermittent connectivity. Interviewees perceived this as a source of disparities, as this issue mainly pertained to rural areas. An interviewee from a rural municipality in Colombia described the challenges of building telehealth services, *"Sometimes there are people who have to climb a hill or climb a tree to be able to catch a signal and sometimes the calls are lost."* In Paraguay, an interviewee indicated that service providers serving indigenous communities did not use digital health at all given lack of connectivity and IT equipment.

## Digital health and the COVID-19 pandemic

During 2020, the COVID-19 pandemic disrupted health services and presented access to care barriers to pregnant and postpartum women. We asked respondents in facilities that used ICTs, the extent to which their use changed as a result of the pandemic. Overall, 42% of respondents said that their ICT use did not change since the COVID-19 pandemic, 39% that they used ICTs more than prior to the pandemic, and 20% that they started using ICTs as a result of the pandemic (Fig 3). Honduras (76%) and the Dominican Republic (68%) had the highest proportion of facilities that did not change their ICT use during the pandemic; facilities in Argentina (63%) and Paraguay (46%) were the most likely to report using ICTs more than before COVID-19; and Peru was the country with more facilities indicating using ICTs for the first time (32%). When asked if they would continue using ICTs in the future, the majority of respondents (84%) indicated they would. The proportion that indicated this was marginally larger in the group who used ICTs more than before the pandemic (90%) (S1 Graph. Future use of ICTs among respondents who used them). One interviewee reflected, *"telehealth will continue in part because it gives us greater flexibility, but we still have to address the problem of connectivity in remote districts."*

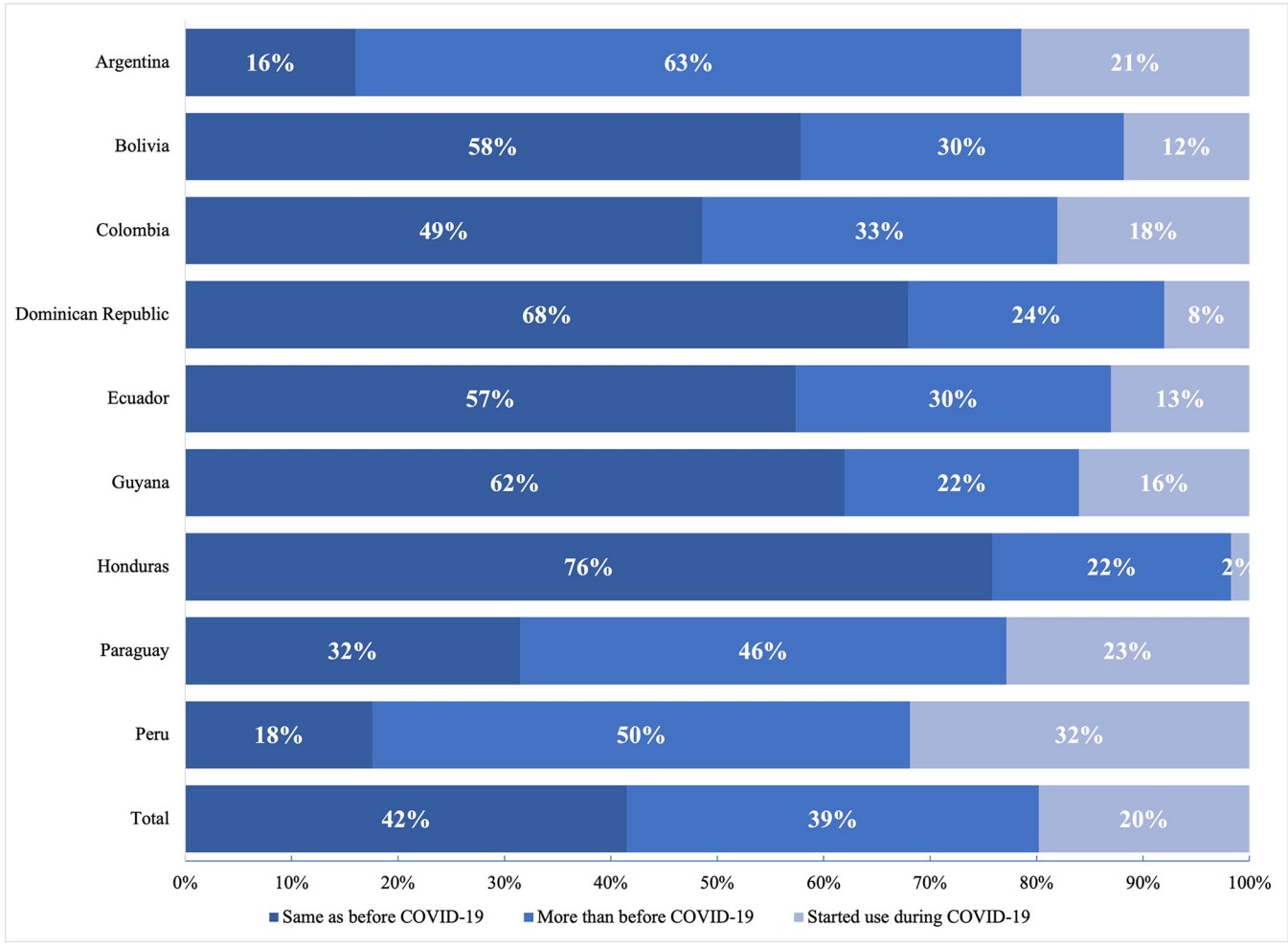

**Fig 3. ICT use in response to COVID-19.**

Qualitative analysis revealed multilevel approaches to minimize COVID-19 infection risk among pregnant women and providers while maintaining continuity of care. At the national level, the Ministry of Health and Social Protection of Colombia issued protocols for the provision of maternity and infant care during the sanitary emergency. The protocols recommended a hybrid model of antenatal visits that involved a combination of antenatal teleconsultations (25%), home visits (25%), and face-to-face consultations (50%) [24]. According to an interviewee, disparities emerged in the implementation of this model. Whereas healthcare providers in the private sector had the infrastructure to provide telehealth consultations, they overused them to reduce costs, not properly evaluating obstetric risks. On the other hand, institutions in the public sector faced challenges in implementing the remote visits, even via telephone, due to a lack of connectivity and equipment. The hybrid antenatal care model was discontinued in 2022.

In Argentina, the ministry of health extended the uses of a telehealth platform originally used solely for teleconsultations between providers to the management and referral of patients with possible COVID-19 infection. The main use of the platform was to evaluate suspected COVID-19 cases and guide the diagnosis and treatment of patients. Teleconsultations were

offered in real-time via videoconferencing. Whereas the platform was not specific to maternity care, pregnant women and recent mothers were considered a priority group.

At the institutional level, a tertiary care maternity hospital in Peru implemented a hybrid antenatal care model for high-risk pregnancies, which entailed alternating consultations and face-to-face visits (50%/50%) [16]. Reflecting on the model's implementation, an administrator noted the importance of an institutional culture of innovation and prior experience in using ICTs to build upon, *"We were the first institution [in the country] to use telehealth. In 2007, we created the maternal-perinatal telehealth network with the institution's own funds, which facilitated interconsultations between our hospital and other facilities throughout the country. . . When in-person consultations closed on March 15, 2020, we began testing a telehealth model."* Interviewees considered the prior use of electronic medical records as a facilitating factor, which enabled providers to access patients' clinical histories from home. Another facilitating factor was having IT personnel on staff to support providers in case of technical difficulties. The hybrid model was widely accepted by users and has expanded since its launch in 2020 to include an app. One limitation of the approach was that it was mainly applied in urban areas of Lima, with good connectivity, in a highly equipped institution, and whose users had access to cell phone or other digital devices.

Another institutional example was a maternity hospital in Argentina which began offering its once in-person birth preparation classes virtually via Zoom, while continuing the normal in-person schedule of antenatal care visits. The modality was highly feasible and acceptable to users. Further, the approach resulted in more family members participating compared to the traditional face-to-face model.

To reduce crowding, facilities in several countries began scheduling antenatal care visits with a set day and time by phone. Prior to the pandemic, providers would only give women a day or week for a return visit. However, interviewees noted that scheduling in advance was difficulted by having physical rather than digital agendas, which made it hard to coordinate scheduling across providers, by women not coming to the scheduled visit, by difficulties anticipating how many patients would come in person without an appointment, and by problems rescheduling in case of an unexpected occurrence (e.g., if the provider is unavailable). Further, some facilities that instructed women to make appointments by phone, did not have enough human resources to answer the phone. As a result, users spent long hours on the phone trying to schedule or reschedule appointments.

In other cases, ICT use was adopted informally by individuals. In institutions across several countries, nurse midwives began calling their patients to check on them. An interviewee from Argentina explained, *"[Concerned with the lack of controls because of the pandemic. . .], and having the perinatal clinical history available, midwives began offering teleconsultations to pregnant women."* These informal approaches, however, were used temporarily in response to the emergency and were largely discontinued.

### National digital and economic setting

To better understand the outer setting in which ICT interventions were implemented in maternity care, this section presents data on economic markers and the national digital infrastructure, including digital coverage, penetration, and affordability. As can be seen in Table 4, all the countries considered in this study are middle-income economies, with Bolivia and Honduras categorized as lower-middle-income and the other countries as upper-middle-income. In terms of digital coverage, most of the population in the countries under study are covered by basic mobile-cellular network. However, gaps remain, particularly in rural areas, with more than 10% of the population not covered by 3G mobile-cellular network in Bolivia, Honduras,

**Table 4. National digital and economic setting, 2021.**

| Country | Infrastructure & Access | | | | | Affordability | | Country Income |
|---|---|---|---|---|---|---|---|---|
| | Network Coverage | | % Population using the Internet | ICTs at Home | | Mobile high consumption basket as % of GNI | Mobile voice and SMS (no data) basket as % of GNI | |
| | Population covered by mobile network | Population covered by 3G mobile network | | Internet access at home | Computer at home | | | |
| Argentina | 98% | 98% | 87% | 90% | 64% | 4.0% | 2.1% | Upper-middle |
| Bolivia | 100% | 88% | 66% | 57% | 28% | 7.9% | 3.6% | Lower-middle |
| Colombia | 100% | 100% | 73% | 61% | 38% | 1.9% | 1.1% | Upper-middle |
| Dominican Republic | 100% | 99% | 85% | 46% | 43% | 4.4% | 1.6% | Upper-middle |
| Ecuador | 96% | 95% | 70% | 60% | 40% | 3.2% | 2.8% | Upper-middle |
| Guyana | 97% | 93% | 85% | | 41% | 5.7% | 2.8% | Upper-middle |
| Honduras | 89% | 82% | 48% | | 16% | 10.5% | 7.2% | Lower-middle |
| Paraguay | 99% | 95% | 77% | 45% | 28% | 3.0% | 3.0% | Upper-middle |
| Peru | 89% | 87% | 71% | 49% | 34% | 1.7% | 1.7% | Upper-middle |
| Regional | 96% | 92% | 81% | 75% | 60% | 3.0% | 2.0% | |

Source of digital environment data: International Telecommunication Union (ITU); Country income based on the World Bank classification. For Infrastructure and Access, the values below the regional average are marked in red; for Affordability values above 2%, the global affordability threshold, are marked in red.

and Peru. As described above, interviewees in many hard-to-reach areas noted that internet connection was unstable or difficult to access. When examining internet use, we observe large differences by country, with the percentage of the population using the internet ranging from 87% in Argentina to 48% in Honduras. In terms of access to ICTs at home, with the exception of Argentina where the government has actively sought to universalize access by distributing personal computers to public school children, a majority of Latin American households do not have home computers. Internet at home is primarily accessed through mobile phones.

Mobile-broadband subscriptions may include three distinct services: voice, SMS messaging and data. Table 4 presents the monthly cost (expressed as % of per capita GNI) of a basic voice and SMS only (no data) basket and of a high-consumption bundle with an allowance of up to 140 minutes of calls, 70 SMS messages, and 2 GB of data. Many of the countries under study lag behind in affordability, with Colombia and Peru being the only countries where the cost of subscriptions is below the 2% affordability threshold.

## Discussion

This mixed methods study was conducted in nine middle-income Latin American and Caribbean countries to understand how, primarily public sector, facilities used ICTs to provide maternity care. Because it was conducted following the COVID-19 pandemic, we were able to examine the extent to which digital health interventions were implemented in response to the sanitary crisis. We found that ICTs were used for various services along the continuum of antenatal to postnatal care and differed across countries and contexts. Overall, 4 out of 5 institutions reported incorporating ICTs in some way in maternity care. The widespread use of

digital health is not surprising given the region's commitment to advancing universal health coverage through technology [25]. Of note, we also found that a digital divide remains, with some populations still unable to benefit from technological developments.

The most frequent uses of ICTs were to provide family planning and breastfeeding counseling over the phone, a largely available technology in the region. These findings are in line with those of a global survey of maternity health professionals, which found that among providers using ICTs, common uses included routine antenatal care (65%), child preparation classes (59%), breastfeeding counseling (50%), and family planning counseling (40%) [12].

Consistent with the literature [26, 27], COVID-19 seems to have accelerated the use of digital health. We found that one-fifth of facilities that used ICTs did so for the first time during the pandemic. Factors associated with the use and adoption of ICTs in maternity care can be classified into individual or team, institutional, and external-level factors, as described below.

## Individual and health team-level factors

We found a wide range of attitudes related to ICT use among providers, from reticence to change to willingness to adopt new technologies and modify existing protocols. The barriers to access created by COVID-19 acted as an incentive for individual providers and health teams to embrace change. In fact, we documented several instances in which maternity care providers initiated the use of ICTs to reach pregnant women during COVID-19. Their initiative, motivation, and creativity were critical to the use of ICTs. Often, the reticence to ICT use was related to low perceived self-efficacy and lack of skills to use new technologies. Indeed, provider training in the use of ICTs was ranked as the leading facilitating factor for their adoption. The adoption of ICTs during the COVID-19 pandemic occurred within a high-stress context, with healthcare providers experiencing burnout, uncertainty, and fear. Learning new processes and acquiring new skills is time-consuming, time which providers did not have during the sanitary emergency.

## Institutional-level factors

While the contributions of individual and health teams were important, perhaps the determining institutional factor affecting the use of ICTs was the information technology infrastructure. Different strategies require different structures, but at a basic level, inadequate telecommunication systems, including reliable internet connection and mobile network signal, was a major barrier to the adoption of ICTs. Lack of ICT equipment, including computers or tables, as needed, was another barrier.

We also found that, often, the workflows, processes, and staffing structures were not compatible with the use of ICTs. Using ICTs requires adequate staffing for providers to have capacity to care for both remote and in person patients, and for support staff, including nursing and administrative staff, to arrange scheduling and pull up patient records. Finally, supervisors should be available to provide guidance and feedback on telehealth provision; and systems must be in place to count telehealth services towards the provider's productivity. In the context of scarce resources, providing basic health services was the priority, with limited opportunities to allocate the human, time, and financial resources to fully embrace an innovation.

## National-level factors

Ultimately, national-level factors were the most important determinants of ICT use. The adoption of digital health at the facility level depended largely on preexisting country commitments to achieving universal health coverage and building the digital infrastructure, evidenced in nearly universal broadband coverage, existing digital healthcare platforms, and public sector

facilities with IT equipment. Facilities in countries with a history of investing in developing the digital infrastructure to ensure access to the entire population and that had begun to build national digital health platforms faced fewer barriers to the adoption of ICTs in maternity care than those without. For example, to carry out teleconsultations between providers in different institutions, interoperable systems to share patient records securely, including laboratory results and imaging, must be in place. Among institutions surveyed, most of the funding to expand ICT use was public and received during the pandemic, highlighting the importance of government investments. Overall, our findings are consistent with prior surveys, that identified the national digital infrastructure and government support for telehealth programs, including public funding, as the top factors determining their use [6].

Another factor impacting the adoption of ICTs was the lack of a policy framework to regulate the use of ICTs, particularly of direct provider-patient care, in many Latin American and Caribbean countries at the beginning of the pandemic [28]. Whereas policy regulations around telehealth lagged behind practice, several governments issued new policies to regulate telehealth during COVID-19 [29], including mandates to public health insurances to reimburse remote care provision.

## Strengths and limitations

This study's findings should be considered in the light of some limitations. Despite efforts to obtain a representative sample of facilities in each county, it should be noted that facilities in distant areas and areas with low internet connectivity were harder to reach. This bias is particularly pertinent to the subject of this analysis, as we may have over-represented the use of ICTs in maternal health. Further, response rates varied by country, and were particularly low in the Dominican Republic. To mitigate this bias, national consultants travelled to hard-to-reach areas to administer the survey in person and/or called potential respondents to administer the survey verbally. Through our extensive outreach, we were able to obtain a large number of responses, making our findings robust. Because we limited the sample to facilities that provided antenatal care, a majority of institutions represented were primary care institutions. A sample with a larger representation of tertiary care facilities may have yielded different findings, such as more frequent use of complex technologies to provide care. The survey was validated with healthcare providers in each country. However, there may have been differences among respondents in the interpretation of the survey questions, thus introducing measurement errors. The mixed methods study focused on the experiences of healthcare providers and administrators primarily in public sector facilities that serve the highest-need populations. Future research could focus on user experiences, such as the satisfaction, acceptability, and cultural appropriateness of using ICTs to receive maternity care. Another limitation is the cross-sectional design, which captured the experiences of facilities at a specific moment in time. The responses illustrated how ICTs were used at the time of the data collection and the changes in their use during the pandemic. However, it was not clear how widely these had been used prior to the pandemic or the extent to which they will continue to be used after the resolution of the sanitary crisis. Future longitudinal research would provide valuable insights into the continuity and evolution of ICT use in maternity care beyond the immediate context of the pandemic. Such research could assess the long-term benefits and challenges associated with telehealth, shedding light on its effectiveness, sustainability, and impact. Despite these limitations, this study is the first to collect data on ICT use in maternity care from institutions in Latin America and the Caribbean at such a large scale. Data from this study contribute to our understanding of the prevalence of ICT use, the technologies used, the services provided, and the factors affecting their use, and can inform the design of interventions to improve maternity care through the integration of digital health solutions.

### Implications for practice

The integration of ICTs in maternity care is a promising strategy in Latin America and the Caribbean. Considerations for their adoption include the level of complexity of the intervention and the cost and availability of the technology required to implement it. Other considerations include the facility's staffing levels, the digital infrastructure, and institutional workflows and protocols, as well as the existing national regulatory framework. Many of the interventions we found relied on the phone, a widely available technology that many people know how to use. However, implementing any innovation requires substantial planning, coordination, and training. In our study, interviewees reported that apparently simple interventions faced challenges. For example, facilities trying to implement advanced scheduling of antenatal care visits by phone encountered insufficient human resources to schedule and/or reschedule visits, and to coordinate care given no-shows and patients without appointments. The more complex the system or platform, the more resources and time are needed for its development and implementation. Health data sharing, for instance, requires systems that are secure and interoperable across institutions.

For facilities ready to expand the use of telehealth, PAHO has developed a tool for institutions to self-assess their readiness to use ICTs, identify gaps, and prioritize investments in improving data use and data collection [30]. The tool provides guidance on assessing telehealth readiness of the infrastructure, human resources and capacity, and the use health data to improve care quality. Further, WHO has also issued guidelines for facilities seeking to incorporate telehealth for systems strengthening [31].

## Conclusions

In this mixed methods study, we found that many facilities throughout the Latin American and Caribbean region incorporated some form of ICTs in maternity care. Respondents considered ICTs as a promising tool to improve care quality and access. During the COVID-19 pandemic, many facilities turned to ICTs to connect with pregnant women, communicate with other providers, and manage patient flow to reduce congestion. Facilities relied on low-cost technologies that were readily available. However, findings also highlighted that ICTs may exacerbate disparities in maternity care. In particular, primary care facilities in rural areas with deficient digital infrastructure were disproportionately excluded from the benefits of telehealth.

Furthermore, the study findings revealed that while there were examples of bilateral investments to develop complex digital platforms, they were not sustainable in the long run. Instead, systems led and coordinated by the government, as part of concerted national efforts to expand the use of ICTs in healthcare, alongside the development of the national digital infrastructure, were found to be the most sustainable and promising for the future.

Overall, more research is needed to understand which digital strategies will persist post-COVID and their impact on access to care and maternal and perinatal outcomes. Research focused on understanding implementation aspects of digital strategies will be particularly useful in promoting scale-up and increasing uptake. Examining the long-term use and outcomes of these digital strategies will provide valuable insights into their effectiveness and potential for addressing existing health disparities.

## Supporting information

**S1 Table. Sampling frame by country.**
(DOCX)

**S2 Table. Response rate by country.**
(DOCX)

**S3 Table. Use of ICTs for abortion counseling by country.**
(DOCX)

**S1 Text. Interview guide.**
(DOCX)

**S2 Text. Online survey questionnaire.**
(PDF)

**S1 Graph. Future use of ICTs among respondents who used them (n = 1459).**
(DOCX)

## Acknowledgments

The authors wish to thank the following individuals for their support in outreach and national data collection: Andrea Mirella Arpita Rojas, Carlos Ayala, María Victoria Bertolino, Alana Browne, Guillermo Carroli, Evelyne Ancion Degraff, Alexia Escóbar Vásquez, Xiomara Fernández, Thais Forster, Rosalinda Hernández, Elodia Jara, Haydée Padilla, Margarita Pérez, Bertha Pooley, Kenia Mariel Ramos Aguilar, Nora Redondo, Catherina Rodriguez, Mirian Rojas, Erick Rousselin, Claudio Sosa, Luis Urbina, Nora Edith Valdivia Soto, and Janice Woolford, as well as all the interviewees in ministries of health and health facilities for their time and invaluable contributions.

## Author Contributions

**Conceptualization:** Ariadna Capasso, Bremen de Mucio.

**Formal analysis:** Ariadna Capasso.

**Funding acquisition:** Suzanne Serruya.

**Investigation:** Ariadna Capasso, Dora Ramírez.

**Methodology:** Ariadna Capasso, Mercedes Colomar, Bremen de Mucio.

**Supervision:** Ariadna Capasso, Bremen de Mucio.

**Writing – original draft:** Ariadna Capasso.

**Writing – review & editing:** Mercedes Colomar, Dora Ramírez, Suzanne Serruya, Bremen de Mucio.

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
