## [Decision Letter · Decision Letter 0]

20 Nov 2023

PONE-D-23-21638Digital health and the promise of equity in maternity care: A mixed methods multi-country assessment on the use of information and communication technologies in healthcare facilities in Latin America and the CaribbeanPLOS ONE

Dear Dr. Capasso,

Thank you for submitting your manuscript to PLOS ONE. After careful consideration, we feel that it has merit but does not fully meet PLOS ONE’s publication criteria as it currently stands. Therefore, we invite you to submit a revised version of the manuscript that addresses the points raised during the review process.

We look forward to receiving your revised manuscript.

Kind regards,

Najmul Hasan, PhD

Academic Editor

PLOS ONE

Journal Requirements:

"This study was partially funded by Global Affairs Canada through an agreement with the Pan American Health Organization/the World Health Organization. Global Affairs Canada was not involved in the study design, collection, analysis, and interpretation of data, writing, and submission."

Reviewers' comments:

Reviewer's Responses to Questions

**Comments to the Author**

1. Is the manuscript technically sound, and do the data support the conclusions?

Reviewer #1: Yes

Reviewer #2: Partly

2. Has the statistical analysis been performed appropriately and rigorously? 

Reviewer #1: Yes

Reviewer #2: No

3. Have the authors made all data underlying the findings in their manuscript fully available?

Reviewer #1: Yes

Reviewer #2: Yes

4. Is the manuscript presented in an intelligible fashion and written in standard English?

Reviewer #1: Yes

Reviewer #2: No

5. Review Comments to the Author

Reviewer #1: The manuscript described a very interesting issue in health area and approached a promissing issue in the present and future time. The main objective is describing the use of information and communication technology in maternal care assistence in nine Latin American countries.

There are some questions to the authors, which are:

-Which was the criteria to select these countries? Why was Brazil not included, as it is one of the biggest countries in Latin America?

-Is there any specific reason to explain the difference of response rate among countries, beyond internet affordability?

-Is there a difference in the use of ICT between the public and private service? What is the percentage of public and private services in each country?

-Page 8: The authors described that “in 7 countries, interviews were conducted with maternal health...”. The question is: in the other 2 countries, with whom the interview was conducted?

-In table 1 (page 12 and 13): why the authors included the column “Used ICTs (%by column)”? I suggest to exclude this column, as there are marked differences between the categories of each item (institution type, level of care and country).

-Review the numering of tables – there are 2 tables numbered as “1”.

Reviewer #2: This is a good article including a massive survey conducted in the region. Please consider the following comments, that I structured in general and specific:

General comments

1. I didn’t understand if the NYU IRB actually reviewed your submission and said it didn’t need ethics approval, or if this was the general instruction that appear in its website. It does look that the informed consents that were used should have been approved by an ethics board.

2. I suggest the authors might review the English, and perhaps asking a native speaker to take a quick look at it. There are several examples in which the English could be improved (e.g., “maternal mortality is a grave public health”).

3. The research questions and how the methods address them are a bit unclear from my perspective. RQ1 and 3 seems to be served by the survey and interviews, but it is not clear how RQ2 would be served by both methods. A

4. Additionally, it is not clear how the methods would be providing complementary perspectives to address the same questions (e.g., what is the difference of addressing RQ1 using the survey as opposed to using the interview results).

5. It would be important to specify a bit more RQ2, and what is understood by “shape the implementation”

6. I was completely confused when the CFIR part came in the methods, and the results/discussion. If you think you could structure the whole research (including the questionnaire, interview guide and key findings) using CFIR, then please do so. But how it is now, it seems a different paper, because it is not clear what research question is being answered with that analysis.

7. By addressing the previous comment, you could also reduce the length of the discussion section which is too long for a paper.

8. In the classification of institutions type, I am not sure about the difference between them (e.g., is a referral hospital not a health center?)

9. One thing that wasn’t clear to me is how use of ICTs is measured. It might be important to clearly state that in the methods section as well as reflecting around this in the discussion section.

10. I got confused in page 24 for the “institutional factors and strategies”. I don’t understand how they are not a subset of factors that are explainednin figure 1?

Specific comments

Introduction

1. Not sure if eradicating maternal deaths is actually an objective. I tend to believe that the verb eradicating is normally used with diseases. Maybe it might be more relevant to use reducing maternal deaths.

2. At the end of introduction, the COVID-19 pandemic is introduced at the end of this section, but seems very disconnected from the rest of the section.

3. It is important the the introducxtion could justify the research questions included. Reading the RQ1, it seems that could have been already answered by reference 14. What additional value is being added by this research piece by answering that research question?

4. Additionally, authors should provide a more robust explanation of the data integration part of the mixed-methods. Are you using triangulation, what type of strategy?

5. It would be important the the authors describe why these nine countries were selected, and why only in 7 were the interviews conducted.

6. Reference 13 is not a review, as it is mentioned in page 6 line 124.

Methods

1. I suggest the authors use clear subheadings following participants selection, data collection methods and data analysis, and clearly splitting the two parts of their study.

2. Also, I believe this is a mixed-methods study that includes a cross-sectional study as a quantatitve component.

3. Page 8, line 159. Please use another paragraph when “in 7 countries” starts.

4. Could you please explain why you had duplicates and a confidence interval for the response rate?

5. Page 9, lines 190-192, all these type of information I am inclined to say that it is part of the results.

Results

1. Page 11 line 238: There is an error.

2. I would suggest that one key issue mentioned is that 78% of the institutions were primary care, meaning that we don’t have much information about the uses of digital health in hospital.

3. Table 1 title: Please specify that these are the results from the survey.

4. In table 1, I think the % by row is confusing because this is not a representative sample.

5. Table numbers are wrong.

6. In table 2, it might be good to have more details on the intersection between the 2 variables shown (e.g., what type of ICT was used for gestational disease management?).

7. Page 17, line 314-315. I think this is should be included in the methods

8. Table 3, could you also provide frequencies?

Discussion

1. I didn’t see strengths, while your survey was massive!

2. I didn’t see implications for practice either.

6. PLOS authors have the option to publish the peer review history of their article (what does this mean?). If published, this will include your full peer review and any attached files.

Reviewer #1: **Yes: **Ana Elisa Madalena Rinaldi

Reviewer #2: **Yes: **Cristian Mansilla

---

## [Author Response · Author response to Decision Letter 0]

14 Dec 2023

PONE-D-23-21638

Digital health and the promise of equity in maternity care: A mixed methods multi-country assessment on the use of information and communication technologies in healthcare facilities in Latin America and the Caribbean

Point-by-point response to reviewers

Journal Requirements

RESPONSE: We modified the manuscript in accordance to PLOS ONE’s style requirements. 

"This study was partially funded by Global Affairs Canada through an agreement with the Pan American Health Organization/the World Health Organization. Global Affairs Canada was not involved in the study design, collection, analysis, and interpretation of data, writing, and submission."

RESPONSE: We included an amended statement in the cover letter, where we also confirm that no additional funding was received for this study.

RESPONSE: The minimal data set has been uploaded to the OpenICPSR repository, and can be found in repository number: https://doi.org/10.3886/E195667V1.

RESPONSE: Separate supporting information files have been uploaded for each element, the in-text citations have been modified to meet PLOS ONE formatting standards, and a list of captions have been included at the end of the manuscript file.

Review Comments to the Author

Please note, the line numbers listed below are those in the revised manuscript with tracked changes document.

5. Review Comments to the Author

Reviewer #1: The manuscript described a very interesting issue in health area and approached a promissing issue in the present and future time. The main objective is describing the use of information and communication technology in maternal care assistence in nine Latin American countries.

There are some questions to the authors, which are:

-Which was the criteria to select these countries? Why was Brazil not included, as it is one of the biggest countries in Latin America?

RESPONSE: This is an important point as Brazil is the country with the highest absolute number of maternal deaths in the region. However, this study was part of a Pan American Health Organization program to improve access to primary care in Latin America and the Caribbean and only included the countries in which this program was implemented. 

We addressed this in the methods (new text in italics):

“This is a cross-sectional mixed methods study of health facilities providing maternity care in 9 Latin American and Caribbean countries that participated in a PAHO-led program to improve primary care access” (lines 163-165)

-Is there any specific reason to explain the difference of response rate among countries, beyond internet affordability?

RESPONSE: Difference in response rates among countries probably depends on the survey dissemination strategy, as well as on internet coverage and the accessibility of equipment. I think the manuscript makes the case that a State that has historically invested in developing a digital infrastructure - as well as a robust public healthcare system – is key to ICT use, in general.

-Is there a difference in the use of ICT between the public and private service? What is the percentage of public and private services in each country?

RESPONSE: This is a great question. Private services probably do have a more developed ICT infrastructure because they have more resources. For example, in Peru, members of EsSalud, which is the government’s paid health insurance for employed citizens, have more options to access telemedicine, than members of SIS, which is the government’s free health insurance. This manuscript primarily focused on free public services for the most vulnerable populations who face the greatest difficulties accessing health care. Each country’s health care systems is pretty fragmented. Determining the percentage of public versus public services available in each would be beyond the scope of this manuscript. However, it would be an interesting future study.

-Page 8: The authors described that “in 7 countries, interviews were conducted with maternal health...”. The question is: in the other 2 countries, with whom the interview was conducted?

RESPONSE: This is a great question. We did not conduct qualitative data collection in two countries. We only conducted interviews in countries where we learned about programs in maternity care that used ICTs, so that we could gather more details about these.

We clarified this point in line 238 (added text in italics):

“In the 7 countries where we found promising examples of use of ICTs in maternity care…”

-In table 1 (page 12 and 13): why the authors included the column “Used ICTs (%by column)”? I suggest to exclude this column, as there are marked differences between the categories of each item (institution type, level of care and country).

RESPONSE: Thank you for this great suggestion. We removed the column.

-Review the numering of tables – there are 2 tables numbered as “1”.

RESPONSE: We were referring to Table 1 twice, the first time to describe the types of institutions that responded to the survey and the other to describe the prevalence of ICT use. Since we modified the table and added a figure (new Fig 1), this is no longer an issue.

Reviewer #2: This is a good article including a massive survey conducted in the region. Please consider the following comments, that I structured in general and specific:

General comments

1. I didn’t understand if the NYU IRB actually reviewed your submission and said it didn’t need ethics approval, or if this was the general instruction that appear in its website. It does look that the informed consents that were used should have been approved by an ethics board.

RESPONSE: Thank you for this point. As per NYU publicly-available ethics committee guidelines, this study did not require submission to the IRB as it did not constitute human subjects’ research. The information collected is not about individuals, but about organizations. The NYU IRB guidance is found here: Does Your Project Require an Application to the NYU IRB Office? Decision Tree #1. Available at: https://www.nyu.edu/content/dam/nyu/research/documents/IRB/IRBDecisionTree.pdf. Even though it didn’t required IRB review, we followed ethical standards by using the consent to inform participants about how the information would be used.

2. I suggest the authors might review the English, and perhaps asking a native speaker to take a quick look at it. There are several examples in which the English could be improved (e.g., “maternal mortality is a grave public health”).

RESPONSE: Thank you for your feedback. We have carefully copy edited the manuscript to ensure that there are no typos or grammatical errors.

3. The research questions and how the methods address them are a bit unclear from my perspective. RQ1 and 3 seems to be served by the survey and interviews, but it is not clear how RQ2 would be served by both methods. 

RESPONSE: We included multiple questions both in the survey and interviews about the context in which ICTs were adopted, including when they were funded and adopted, among others, to help us understand if and how changes in ICT use occurred during COVID-19.

4. Additionally, it is not clear how the methods would be providing complementary perspectives to address the same questions (e.g., what is the difference of addressing RQ1 using the survey as opposed to using the interview results).

RESPONSE: The methods are complementary because whereas the survey allowed us to reach more institutions across the region and quantify the use of ICTs in the 9 countries assessed, the qualitative data allowed us to gather more nuanced information related to the impetus for ICT adoption, the context in which the adoption of ICTs occurred, the exact use of ICTs (impetus, team involved, intervention characteristics, etc.).

We explained this in lines 248-251:

“The information gathered from the different data collection methods was complementary: whereas the survey allowed us to collect quantitative data from many institutions, the qualitative data provided us nuanced information on the context in which specific interventions were implemented.”

5. It would be important to specify a bit more RQ2, and what is understood by “shape the implementation”

RESPONSE: We appreciate the point and have modified the language in RQ2 to address this point (rephrased RQ2 in italics, line 156):

1) Understand changes in the use of ICTs as a result of the COVID-19 pandemic

6. I was completely confused when the CFIR part came in the methods, and the results/discussion. If you think you could structure the whole research (including the questionnaire, interview guide and key findings) using CFIR, then please do so. But how it is now, it seems a different paper, because it is not clear what research question is being answered with that analysis.

RESPONSE: We made significant changes to the introduction and discussion to address this issue; we reorganized the discussion and removed the sections related to the CFIR framework. Whereas we still believe that CFIR provides a useful framework to understand the adoption of innovations in a real-world setting, we agree that this content would best suit a separate paper.

7. By addressing the previous comment, you could also reduce the length of the discussion section which is too long for a paper.

RESPONSE: Thank you for the suggestion. Indeed, the manuscript is much shorter with the removal of the CFIR sections.

8. In the classification of institutions type, I am not sure about the difference between them (e.g., is a referral hospital not a health center?)

RESPONSE: A health center is a health facility that only provides primary level care, as compared to a hospital which can offer higher complexity health services. To avoid confusion, we clarified this point by changing the terminology to “primary care health center.”

9. One thing that wasn’t clear to me is how use of ICTs is measured. It might be important to clearly state that in the methods section as well as reflecting around this in the discussion section.

RESPONSE: ICT use was measured by the following survey question:

Does your facility use any information and communication technology (ICT) tools for maternal health care (e.g., phone calls, video calls, telemonitoring, text messaging, etc.)?

The answer options were: Yes, the same way as we did before the onset of the pandemic (1); Yes, more than before the pandemic (2); Yes, we started using ICT tools when the pandemic was already under way (3); and No (4).

We added this information to the methods:

“ICT use was assessed by the question “Does your facility use any information and communication technology (ICT) tools for maternal health care (e.g., phone calls, video calls, telemonitoring, text messaging, etc.)?” (Answer options: Yes, the same way as we did before the onset of the pandemic; Yes, more than before the pandemic; Yes, we started using ICT tools when the pandemic was already under way; and No). The variable was dichotomized Yes/No.” (lines 215-219)

10. I got confused in page 24 for the “institutional factors and strategies”. I don’t understand how they are not a subset of factors that are explainednin figure 1?

RESPONSE: This confusion is understandable. Figure 1 presented survey respondents’ perceptions of the factors that facilitated ICT use, whereas Table 3 are the actual strategies implemented by institutions to facilitate ICT use. To avoid confusion, we incorporated findings from the “institutional factors and strategies” section into the factors associated with ICT use.

Specific comments

Introduction

1. Not sure if eradicating maternal deaths is actually an objective. I tend to believe that the verb eradicating is normally used with diseases. Maybe it might be more relevant to use reducing maternal deaths.

RESPONSE: Thank you. We agree with the comment and changed the language to reducing. (line 67)

2. At the end of introduction, the COVID-19 pandemic is introduced at the end of this section, but seems very disconnected from the rest of the section.

RESPONSE: We appreciate this suggestion. We wove in the introduction to COVID-19 within the section presenting prior evidence on the use of ICTs in the region. The main point we were trying to make was that by creating barriers to accessing services in person, COVID-19 may have accelerated the adoption of ICTs. Thus, warranting this study on the use of ICTs in the post-pandemic period (lines 121-125).

3. It is important the the introducxtion could justify the research questions included. Reading the RQ1, it seems that could have been already answered by reference 14. What additional value is being added by this research piece by answering that research question?

RESPONSE: Reference 14 is a desk review of published evidence on the use of ICTs in sexual and reproductive health. However, the vast majority of health interventions carried out by health facilities in Latin America and the Caribbean are not documented. Thus, any study based solely on desk review is vastly limited. Primary data collection via survey and interviews provides a more robust picture of the situation of ICT use in the region.

We clarified this in the manuscript:

“A caveat of this study is that, since the majority of interventions carried out in facilities across Latin America and the Caribbean are not documented in the literature, a desk review provides only a partial picture of the reality on the ground..” (lines 138-141)

4. Additionally, authors should provide a more robust explanation of the data integration part of the mixed-methods. Are you using triangulation, what type of strategy?

RESPONSE: The survey findings were used to present quantitative data, primarily related to prevalence of ICT use, types of ICTs used, and services provided. The interviews served to learn more about specific practices and the context in which ICTs were used. As mentioned above, the methods are complementary because whereas the survey allowed us to reach more institutions and quantify the use of ICTs, the qualitative data allowed us to gather more nuanced information related to the impetus for ICT adoption, the context in which the adoption of ICTs occurred, the exact use of ICTs (purposes, technology used, team involved, etc.).

We explained this in lines 248-251:

“The information gathered from the different data collection methods was complementary: whereas the survey allowed us to collect quantitative data from many institutions, the qualitative data provided us nuanced information on the context in which specific interventions were implemented.”

5. It would be important the the authors describe why these nine countries were selected, and why only in 7 were the interviews conducted.

RESPONSE: This is a great point. This study was part of a Pan American Health Organization program to improve access to primary care in Latin America and the Caribbean and only included the countries in which this program was implemented. We only conducted interviews in countries where we learned about programs in maternal health that used ICTs, so that we could learn more about these. 

We addressed this in the methods (new text in italics):

“This is a cross-sectional mixed methods study of health facilities providing maternity care in nine Latin American and Caribbean countries that participated in a PAHO-led program to improve primary care access” (lines 164-165)

“In the 7 countries where we found promising examples of use of ICTs in maternal health care…” (line 238-239)

6. Reference 13 is not a review, as it is mentioned in page 6 line 124.

RESPONSE: Thank you for the observation. We removed that sentence and reworded that paragraph to better reflect the evidentiary sources. (lines 129-134)

Methods

1. I suggest the authors use clear subheadings following participants selection, data collection methods and data analysis, and clearly splitting the two parts of their study.

RESPONSE: Thank you for the suggestion. We separated the quantitative and qualitative descriptions of the study methods into sections.

2. Also, I believe this is a mixed-methods study that includes a cross-sectional study as a quantatitve component.

RESPONSE: This is both a mixed-methods study because it uses quantitative and qualitative data collection methods AND a cross-sectional study because there is no repeating or longitudinal data – data from each resondents was collected at one time point.

3. Page 8, line 159. Please use another paragraph when “in 7 countries” starts.

RESPONSE: This section was moved when we reorganized the methods. We made sure it starts a new paragraph. (line 238)

4. Could you please explain why you had duplicates and a confidence interval for the response rate?

RESPONSE: We had duplicates because some respondents started the online survey and stopped for one reason or another (e.g., their internet was cut, they were interrupted by something else), and then instead of continuing the survey, they restarted it. This only happened when they used a different device to try to complete the survey. We calculated the confidence interval of the mean response rate to account for random variance across countries.

We clarified: “Duplicates occurred when respondents continued working on unsubmitted surveys from a different device.” (lines 275-276)

5. Page 9, lines 190-192, all these type of information I am inclined to say that it is part of the results.

RESPONSE: These lines were moved to the beginning of the results section. (lines 275-279)

Results

1. Page 11 line 238: There is an error.

RESPONSE: Thank you for pointing this out. We changed to the actual percentage, 76%. (line 302)

2. I would suggest that one key issue mentioned is that 78% of the institutions were primary care, meaning that we don’t have much information about the uses of digital health in hospital.

RESPONSE: Thank you. We have included this point in the discussion.

3. Table 1 title: Please specify that these are the results from the survey.

RESPONSE: The title of the table was modified to include that these were the results from the survey (lines 304).

4. In table 1, I think the % by row is confusing because this is not a representative sample.

RESPONSE: Thank you for your suggestion. We removed the percentage by row column.

5. Table numbers are wrong.

RESPONSE: We were referring to Table 1 twice, the first time to describe the types of institutions that responded to the survey and the other to describe the prevalence of ICT use. Since we modified the table and added a figure (new Fig 1), this is no longer an issue.

6. In table 2, it might be good to have more details on the intersection between the 2 variables shown (e.g., what type of ICT was used for gestational disease management?).

RESPONSE: This is a good suggestion but cannot be answered with the current data. We could do crosstabs on the two variables, but that still wouldn’t tell us specifically what types of ICTs were used for a specific condition such as gestational disease management. 

7. Page 17, line 314-315. I think this is should be included in the methods

RESPONSE: Thank you for the suggestion. We moved this to the survey questionnaire section. (lines 215-217)

8. Table 3, could you also provide frequencies?

RESPONSES: Thank you the suggestion. We added the frequencies to Table 3.

Discussion

1. I didn’t see strengths, while your survey was massive!

RESPONSE: Thank you! We added strengths to the Strengths and Limitations subsection (line 780).

2. I didn’t see implications for practice either.

RESPONSE: Valuable point. We added a section on implications for practice in the discussion (starting in line 813).

---

## [Decision Letter · Decision Letter 1]

16 Jan 2024

PONE-D-23-21638R1Digital health and the promise of equity in maternity care: A mixed methods multi-country assessment on the use of information and communication technologies in healthcare facilities in Latin America and the CaribbeanPLOS ONE

Dear Dr. Capasso,

Thank you for submitting your manuscript to PLOS ONE. After careful consideration, we feel that it has merit but does not fully meet PLOS ONE’s publication criteria as it currently stands. Therefore, we invite you to submit a revised version of the manuscript that addresses the points raised during the review process.

We look forward to receiving your revised manuscript.

Kind regards,

Najmul Hasan, PhD

Academic Editor

PLOS ONE

Journal Requirements:

Reviewers' comments:

Reviewer's Responses to Questions

**Comments to the Author**

1. If the authors have adequately addressed your comments raised in a previous round of review and you feel that this manuscript is now acceptable for publication, you may indicate that here to bypass the “Comments to the Author” section, enter your conflict of interest statement in the “Confidential to Editor” section, and submit your "Accept" recommendation.

Reviewer #2: (No Response)

Reviewer #3: All comments have been addressed

2. Is the manuscript technically sound, and do the data support the conclusions?

Reviewer #2: Yes

Reviewer #3: Yes

3. Has the statistical analysis been performed appropriately and rigorously? 

Reviewer #2: Yes

Reviewer #3: Yes

4. Have the authors made all data underlying the findings in their manuscript fully available?

Reviewer #2: Yes

Reviewer #3: Yes

5. Is the manuscript presented in an intelligible fashion and written in standard English?

Reviewer #2: Yes

Reviewer #3: Yes

6. Review Comments to the Author

Reviewer #2: Thank you for addressing all the previous comments. I think most of them were already addressed. I only have 2 follow-up points. I add my original point + your response in each:

4. Additionally, it is not clear how the methods would be providing complementary

perspectives to address the same questions (e.g., what is the difference of addressing

RQ1 using the survey as opposed to using the interview results).

RESPONSE: The methods are complementary because whereas the survey allowed

us to reach more institutions across the region and quantify the use of ICTs in the 9

countries assessed, the qualitative data allowed us to gather more nuanced

information related to the impetus for ICT adoption, the context in which the adoption of

ICTs occurred, the exact use of ICTs (impetus, team involved, intervention

characteristics, etc.).

We explained this in lines 248-251:

“The information gathered from the different data collection methods was

complementary: whereas the survey allowed us to collect quantitative data from many

institutions, the qualitative data provided us nuanced information on the context in

which specific interventions were implemented."

4. Additionally, authors should provide a more robust explanation of the data

integration part of the mixed-methods. Are you using triangulation, what type of

strategy?

RESPONSE: The survey findings were used to present quantitative data, primarily

related to prevalence of ICT use, types of ICTs used, and services provided. The

interviews served to learn more about specific practices and the context in which ICTs

were used. As mentioned above, the methods are complementary because whereas

the survey allowed us to reach more institutions and quantify the use of ICTs, the

qualitative data allowed us to gather more nuanced information related to the impetus

for ICT adoption, the context in which the adoption of ICTs occurred, the exact use of

ICTs (purposes, technology used, team involved, etc.).

Follow-up: I will connect both points because I think I have the same comment. Thanks for your explanation, but I feel that the complementarity of both methods is still unclear. You explained very clear how the quantitative and qualitative data contributes to answer the question, but it would be helpful to say the survey answered the question related to prevalence, types of ICTs, and services provided, while the qualitative data answered the more nuanced information. I think these should be separate research questions clearly specified at the end of the introduction. I say this because otherwise readers might think that you are answering the same question using different methodologies, and then triangulation or other mixed-methods data integration method is expected.

Just this minor point, but the rest looks good to me!

Reviewer #3: Title: Digital health and the promise of equity in maternity care: A mixed methods multicountry assessment on the use of information and communication technologies in

healthcare facilities in Latin America and the Caribbean

Abstract: Key study aim, methods and result of the study well presented.

Introduction: Detailed information on statement of problem, rational for the study clearly presented and study objectives are well presented.

Methods: Fairly well structured and described. However, the following observation need to be addressed

• Survey sampling - Please provide the inclusion and exclusion criteria for selecting the facilities.

• Was the sampling frame restricted to public health facilities? The sampling technique used for each of the countries are not mentioned.

• Interviewees were maternal health unit directors or telehealth focal points - In facilities where they have both, who is selected for the interview? This is likely to be the case in a number of health facilities used. Is it focal points or head of the telehealth units that were interviewed?

Result: Well written in details with relevant figures – suggestions from previous reviewers were done satisfactorily.

Discussion: The study findings are well discussed, with study limitations provided.

Conclusion: Clearly written with appropriate recommendation.

7. PLOS authors have the option to publish the peer review history of their article (what does this mean?). If published, this will include your full peer review and any attached files.

Reviewer #2: **Yes: **Cristian Mansilla

Reviewer #3: **Yes: **Prof. Tanimola Makanjuola Akande

---

## [Author Response · Author response to Decision Letter 1]

21 Jan 2024

PONE-D-23-21638R1

Digital health and the promise of equity in maternity care: A mixed methods multi-country assessment on the use of information and communication technologies in healthcare facilities in Latin America and the Caribbean

PLOS ONE

Dear Dr. Hasan,

Thank you for the opportunity to resubmit the above-mentioned manuscript. Below my point-by-point response to reviewers.

Best regards,

Ariadna

Journal Requirements:

RESPONSE: We have reviewed the reference list to ensure completeness.

6. Review Comments to the Author

Reviewer #2: Thank you for addressing all the previous comments. I think most of them were already addressed. I only have 2 follow-up points. I add my original point + your response in each:

4. Additionally, it is not clear how the methods would be providing complementary

perspectives to address the same questions (e.g., what is the difference of addressing

RQ1 using the survey as opposed to using the interview results).

RESPONSE: The methods are complementary because whereas the survey allowed

us to reach more institutions across the region and quantify the use of ICTs in the 9

countries assessed, the qualitative data allowed us to gather more nuanced

information related to the impetus for ICT adoption, the context in which the adoption of

ICTs occurred, the exact use of ICTs (impetus, team involved, intervention

characteristics, etc.).

We explained this in lines 248-251:

“The information gathered from the different data collection methods was

complementary: whereas the survey allowed us to collect quantitative data from many

institutions, the qualitative data provided us nuanced information on the context in

which specific interventions were implemented."

4. Additionally, authors should provide a more robust explanation of the data

integration part of the mixed-methods. Are you using triangulation, what type of

strategy?

RESPONSE: The survey findings were used to present quantitative data, primarily

related to prevalence of ICT use, types of ICTs used, and services provided. The

interviews served to learn more about specific practices and the context in which ICTs

were used. As mentioned above, the methods are complementary because whereas

the survey allowed us to reach more institutions and quantify the use of ICTs, the

qualitative data allowed us to gather more nuanced information related to the impetus

for ICT adoption, the context in which the adoption of ICTs occurred, the exact use of

ICTs (purposes, technology used, team involved, etc.).

Follow-up: I will connect both points because I think I have the same comment. Thanks for your explanation, but I feel that the complementarity of both methods is still unclear. You explained very clear how the quantitative and qualitative data contributes to answer the question, but it would be helpful to say the survey answered the question related to prevalence, types of ICTs, and services provided, while the qualitative data answered the more nuanced information. I think these should be separate research questions clearly specified at the end of the introduction. I say this because otherwise readers might think that you are answering the same question using different methodologies, and then triangulation or other mixed-methods data integration method is expected.

Just this minor point, but the rest looks good to me!

RESPONSE: We have addressed this request to clarify how quantitative and qualitative data informed the study by modifying the first study objective, as follows (next text in italics):

Describe the prevalence of ICT use in maternity care, the types of ICTs used, and the services provided using ICTs in a sample of health facilities in 9 countries of Latin America and the Caribbean (page 7)

And be adding the following description in the methods (page 8):

“Survey data served primarily to answer questions related to the prevalence of ICT use, the types of ICTs used, and the services provided using ICTs, whereas qualitative data served to gain nuanced information about the context in which ICTs were employed, and about the barriers and facilitators to their adoption.”

Reviewer #3: Title: Digital health and the promise of equity in maternity care: A mixed methods multicountry assessment on the use of information and communication technologies in

healthcare facilities in Latin America and the Caribbean

Abstract: Key study aim, methods and result of the study well presented.

Introduction: Detailed information on statement of problem, rational for the study clearly presented and study objectives are well presented.

Methods: Fairly well structured and described. However, the following observation need to be addressed

• Survey sampling - Please provide the inclusion and exclusion criteria for selecting the facilities.

RESPONSE: We specified the inclusion and exclusion criteria as follows (page 8_:

“The study included health facilities that provided maternity care in the countries under consideration, and excluded those that did not provide such care.” 

• Was the sampling frame restricted to public health facilities? The sampling technique used for each of the countries are not mentioned.

RESPONSE: This was part of the negotiation with the country teams. Most countries restricted the survey to public health facilities, with the private sector only included in Guyana, Paraguay, and Peru. Column 2 of the S1 Table with the sampling frame was modified to include this detail by country. This detail was also added to the survey sampling description (pages 8-9):

“The country teams decided whether to limit the survey to the public sector or not. Most countries focused on public sector facilities, while Guyana, Paraguay, and Peru included those in both the public and private sectors.”

• Interviewees were maternal health unit directors or telehealth focal points - In facilities where they have both, who is selected for the interview? This is likely to be the case in a number of health facilities used. Is it focal points or head of the telehealth units that were interviewed?

RESPONSE: In addition to facility-level providers, we tried to interview both telehealth and maternal health focal points at the ministries of health. Whether we reached both, depended on the national context, and on who was available and willing to participate.

---

## [Decision Letter · Decision Letter 2]

1 Feb 2024

Digital health and the promise of equity in maternity care: A mixed methods multi-country assessment on the use of information and communication technologies in healthcare facilities in Latin America and the Caribbean

PONE-D-23-21638R2

Dear Dr. Capasso

We’re pleased to inform you that your manuscript has been judged scientifically suitable for publication and will be formally accepted for publication once it meets all outstanding technical requirements.

Kind regards,

Najmul Hasan, PhD

Academic Editor

PLOS ONE

Additional Editor Comments (optional):

Reviewers' comments:

Reviewer's Responses to Questions

**Comments to the Author**

1. If the authors have adequately addressed your comments raised in a previous round of review and you feel that this manuscript is now acceptable for publication, you may indicate that here to bypass the “Comments to the Author” section, enter your conflict of interest statement in the “Confidential to Editor” section, and submit your "Accept" recommendation.

Reviewer #2: All comments have been addressed

Reviewer #3: All comments have been addressed

2. Is the manuscript technically sound, and do the data support the conclusions?

Reviewer #2: Yes

Reviewer #3: (No Response)

3. Has the statistical analysis been performed appropriately and rigorously? 

Reviewer #2: Yes

Reviewer #3: (No Response)

4. Have the authors made all data underlying the findings in their manuscript fully available?

Reviewer #2: Yes

Reviewer #3: (No Response)

5. Is the manuscript presented in an intelligible fashion and written in standard English?

Reviewer #2: Yes

Reviewer #3: (No Response)

6. Review Comments to the Author

Reviewer #2: (No Response)

Reviewer #3: (No Response)

7. PLOS authors have the option to publish the peer review history of their article (what does this mean?). If published, this will include your full peer review and any attached files.

Reviewer #2: **Yes: **Cristián Mansilla

Reviewer #3: **Yes: **Prof. Tanimola Makanjuola AKANDE

---

## [Editor Report · Acceptance letter]

19 Feb 2024

PONE-D-23-21638R2 

PLOS ONE

Dear Dr. Capasso, 

I'm pleased to inform you that your manuscript has been deemed suitable for publication in PLOS ONE. Congratulations! Your manuscript is now being handed over to our production team.

Kind regards, 

on behalf of

Dr. Najmul Hasan 

Academic Editor

PLOS ONE